# Adjusted spectral correction method for calculating extreme winds in tropical cyclone affected water areas

Xiaoli Guo Larsén[1] and Søren Ott[1]

[1]Department of Wind and Energy Systems, Technical University of Denmark

**Correspondence:** Xiaoli Guo Larsén (xgal@dtu.dk)

**Abstract.** A method is developed to calculate the extreme wind for tropical cyclone affected water areas. The method is based on the spectral correction method by Larsén et al. (2012) in connection with the use of numerically modeled data, where an enhancement coefficient is derived as a function of wind speed to reflect the large wind fluctuation during tropical cyclones. This is done through calibration with the estimates from Ott (2005) who used the best track data and Holland model to estimate the extreme wind over the Typhoon affected area in the western North Pacific. The method is applied in the current study to three regions where the 50-year winds with an effective temporal resolution of 10 minutes are obtained at 10 m, 50 m, 100 m and 150 m. The results are in agreement with Ott (2005) over their study domain, though with much more spatial details of the extreme wind distribution.

## 1 Introduction

The 50-year wind at turbine hub height ($U_{50}$) is one of the most important siting parameters that needs to be estimated for the wind turbine design (e.g. IEC, 2019; Yu et al., 2011). To find the correct type of wind turbines for a particular area is a necessary step for regional wind energy planning, as it is related to Levelized Cost of Energy (LCoE), that involves wind resource, safety and risk in connection with design, operation and maintenance.

Wind energy development has been expeditious in the past decade and wind turbines are being raised or planned in many regions globally, including tropical cyclone affected offshore areas.

The most reliable method to estimate $U_{50}$ should be to use good quality measurements at the site that are sufficiently long, e.g. more than 10 years. This is however a condition that is almost never satisfied in these areas. Measurements under hurricane conditions are difficult and expensive to obtain, due mostly to technical difficulties at winds of such strength. A few meteorological stations exist in some tropical cyclone affected regions over coastal land areas. It is far from being enough for assessing the extreme conditions, particularly over water. Moreover, most of these measurements are not accessible for general research purposes.

Alternatively, people use modeling approaches to obtain extreme wind statistics related to tropical cyclones. The often-used modeling approaches include the stochastic Monte Carlo Simulation (MCS) (e.g. IEC, 2019), the simplified physical Hurricane Holland model (Holland, 1980), and numerical weather models.

In this study we focus on physics-based modelling approaches and therefore will not discuss MCS. In using the Holland model, Ott (2005) jointly used the best track data from 1977 to 2004 and calculated $U_{50}$ at 10 m for the region of western North Pacific ocean; his method is referred to as the Ott method here. The values of $U_{50}$ from Ott (2005) are of an equivalent temporal resolution of 10-min. The simple Holland model describes some of the most important parameters of a tropical cyclone, including the minimum center pressure, maximum wind speed and the distance to the cyclone eye. It also assumes a spatially symmetric distribution of wind speed from the cyclone eye to outer region. The simplicity of the Ott method at the same time caused its limitation, e.g. the lack of detailed spacial information of the wind speed, horizontally and vertically. Moreover, the Ott method needs feed-in information of the best track data.

Numerical modeling has been a powerful tool in simulating tropical cyclones, as being done in research and forecasting centers for Hurricanes and Typhoons. These weather models can be run at a spatial resolution from a few kilometers to a few tens of kilometers, with outputs most-often saved hourly. These long-term simulation data are valuable for assessing the extreme wind in tropical cyclone affected areas. However, most of these simulations are forecasts with model schemes and setups that undergo regular and continuous updates and improvements. The updates are advantageous for forecasting purposes, but not necessarily for calculating the extreme winds, because inconsistency is introduced into the long-term data, which will affect the extreme wind samples artificially. Moreover, the data from weather centers are not always accessible for wind energy users.

The reanalysis data thus have been a very attractive option, as they are generated using a consistent setup and they are globally available with open access. Some of them, e.g. Climate Forecast System Reanalysis (CFSR) and the fifth generation ECMWF reanalysis (ERA-5) come with a spatial resolution of about 25 - 40 km, available at hourly basis. For instance, Pryor and Bartelmie (2021) have used the ERA-5 data and created a global atlas of extreme winds at 100 m. However, their validation suggests that the extreme winds in tropical cyclone affected areas are significantly underestimated. This is nevertheless expected, as Larsén et al. (2012) showed through spectral analysis of the modeled wind speed time series. Wind time series from mesoscale numerical models in general suffers from the numerical smoothing effect, introduced by a coarse grid that facilitates the convergence of the model. Larsén et al. (2012) showed that the modeled time series misses the high-frequency variability in comparison with measurements of similar resolution, causing systematic underestimation of the 50-year wind. This is the case almost ever-present in data from mesoscale modeling, and the smoothing effect is expected to be significant for reanalysis data of tens of kilometers spatial resolution. To solve this problem, Larsén et al. (2012) developed a so-called spectral correction (SC) method to fill in the missing wind variability from modeled time series through a spectral model. Thus the corrected time series will follow the power spectrum down to the temporal resolution of the measurements. To meet the IEC standards, we can correct the spectrum to an equivalent resolution of 10 min. This SC method is briefly introduced in section 2.2.1 here.

The SC method has been used in connection with different reanalysis data to predict extreme winds globally, including CFSR and Climate Four-Dimensional Data Assimilation (CFDDA) (Larsén and Kruger, 2014; Hansen et al., 2016; Larsén et al., 2022). The SC method has been shown to be reliable when validated with mid-latitude measurements. However, applying SC in the same way as for mid-latitude storms suggests a significant underestimation of $U_{50}$ for the western north Pacific Ocean, when comparing with the results from Ott (2005).

The current study provides a simple approach for all tropical cyclone affected water areas, combining the strength of the method from Ott (2005) and Larsén et al. (2012). This method is denoted here as the spectral correction for tropical cyclone method, in short, the SC-TC method.

This new method is described in section 2. Section 3 shows results with discussions for the western Pacific Ocean as well as two other regions that are under the impact of tropical cyclones. Summary and conclusions are provided in section 4.

## 2 The development of the method

### 2.1 The data

In this study, in connection with the development of the SC-TC method, we have used the wind speed at 10 m from CFSR-1 reanalysis data. The CFSR-1 data are available from 1979 - 2010, and they are hourly values at a spatial resolution of about 40 km (Saha et al., 2010). There is a version-2 CFSR data, CFSv2, which are available from 2011, with a higher spatial resolution of about 25 km. However, the total data length of CFSv2 is too short for the calculation of the 50-year wind. The CFSv2 data are however, used to investigate the wind speed spectra during a Typhoon case Megi in section 2.2.2. Outputs for Megi from the mesoscale Weather Research and Forecasting (WRF) model, with a spatial resolution of 2 km, are also used.

When developing the SC-TC method, we also used the estimation of the 50-year wind at 10 m from Ott (2005), which is derived from best track data, and Holland model, for an area over the western north Pacific ocean. These values, called here $U_{50,BT}$, where $BT$ stands for Best Track, are available on grid size of about $1°$. They are from Fig. 13 in Ott (2005) and re-produced here as contour lines in Fig. 4a.

### 2.2 The spectral correction method for tropical cyclone conditions

#### 2.2.1 The spectral correction method

As explained in Section 1, the smoothened time series from numerical models results in missing wind variability and, accordingly, translates into low spectral energy at higher frequencies when compared to measurements (Larsén et al., 2012). Fig. 1 shows an example of the spectrum from modeled time series (black curve, from the 32-year CFSR-1 data) in comparison with expected spectral tail that has a slope of $-5/3$ in a log-log coordination (solid red curve).

The effect of the smoothing of the time series on the estimation of the extreme wind is calculated in Larsén et al. (2012) by assuming the time series follows a Gaussian process. The once-per-year exceedance can thus be described using the Poisson distribution. Following this, in Larsén et al. (2012), the wind that occurs once a year $\overline{U}_{\max}$ was derived as a function of the

zeroth- and second-order spectral moments $m_0$ and $m_2$:

$$\overline{U}_{\max} = \overline{U} + \sqrt{m_0}\sqrt{2\ln\left(\sqrt{\frac{m_2}{m_0}}T_0\right)} \tag{1}$$

where $\overline{U}$ is the mean wind speed, $T_0$ is the basis period of one year and $m_i$ is the $i$th spectral moment defined by

$$m_i = 2\int\limits_0^\infty f^i S(f)\,df \tag{2}$$

where $f$ is the frequency in Hz and $S(f)$ is the spectrum of the wind speed. From the above equations, it is clear that $\overline{U}_{\max}$ is significantly affected by the high frequency part of the spectrum through $m_2$. Thus, we can calculate the smoothing effect on $\overline{U}_{\max}$ from the modeled time series by correcting its spectral tail.

Such a correction has been done by replacing the high frequency part of the spectrum from the modeled time series with the following spectral model:

$$S(f) = a\cdot f^{-5/3} \tag{3}$$

from a certain frequency $f_c$ to the expected frequency $f_h$. In our study it is the 10-min temporal resolution we aim at, so that $f_h = 72$ day$^{-1}$ (0.00083 Hz), which is the Nyquist frequency for 10-min. Eq. 3 is the mesoscale part of the expression from Larsén et al. (2013), which includes both a synoptic and mesoscale range of spectrum: $S(f) = a\cdot f^{-5/3} + b\cdot f^{-3}$. As shown by Fig. 1, $a$ is determined by the time series of the model data where the black curve and the red solid curve meet. In Hansen et al. (2016), $f_c = 0.8$ day$^{-1}$ was used. In Larsén et al. (2022), $f_c$ and $S(f_c)$ were chosen from a regression line of $\ln S(f)$ with $\ln f$ for the range $0.6 < f < 0.9$ day$^{-1}$, thus the choice of $f_c$ becomes less sensitive to the fluctuation in $S(f)$.

Using the original spectrum and the corrected spectrum, together with the mean wind speed $\overline{U}$ to Eq. 3, we obtain $\overline{U}_{max,ori}$ and $\overline{U}_{max,SC}$, and their ratio $R = \overline{U}_{max,SC}/\overline{U}_{max,ori}$. We use this ratio $R$ to correct the annual maximum wind speed obtained from the time series directly. From a time series of $N-$year data, we obtain $N$ samples of the annual maximum wind speed. After applying the spectral correction, we use the Annual Maximum Method with the Gumbel distribution to obtain the 50-year wind (e.g. Abild et al., 1992; Larsén et al., 2019).

### 2.2.2   Example of the spectral behaviours during a tropical cyclone

The above-mentioned SC method was shown to provide reasonable estimates for mid-latitude storms (e.g. Hansen et al., 2016; Larsén and Kruger, 2014). However, using Eq. 3 results in underestimation of $U_{50}$ for the tropical cyclone affected areas. It is because the energy level in the mesoscale range during tropical cyclones is significantly higher. This is demonstrated in Fig. 2 through an example of spectral analysis during typhoon case Megi that past Taiwan during the period 26th - 27th Sep. 2016. It is not the focus of current study to investigate mesoscale modeling of the tropical cyclones, thus, details of the simulation of typhoon Megi using the WRF model are given in Appendix A. As the period is too short for a Fourier analysis of the time series in time domain, we examine the power spectra in wavenumber domain, namely, $E(k)$ vs. wavenumber ($k$). We compare

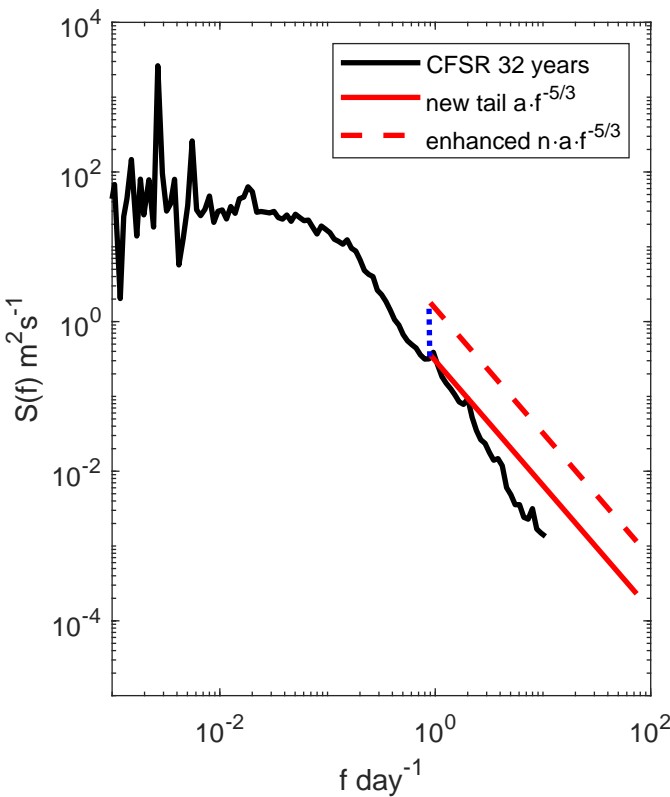

**Figure 1.** Illustration of extending the SC method to the SC-TC method, where the black curve shows the wind speed spectrum from the model data (here, 32-year CFSR-1 wind speed at 10 m), the red solid curve is Eq. 3 and the red dashed curve is Eq. 4. The vertical blue dotted line collects the red curve to the original spectrum for the spectral correction.

the spectral behaviour and energy level for the wind speed between this particular case and climatological conditions in Fig. 2. Spectra $E(k)$ corresponding to Megi were calculated over an area of about 200 km by 200 km over the Pacific Ocean as Megi approached Taiwan during the period 2016-09-26 10:00 to 2016-09-27 09:00; both the reanalysis data CFSv2 (blue in Fig. 2) and the WRF data were used (red). We calculated and averaged the one-dimensional spectra in the North-South (N-S) direction as well as West-East (W-E) direction. They are plotted as solid and dashed curves, respectively, in Fig. 2. Thus, the spectra represent an average energy level over the entire area, which is more representative for this particular case in comparison with a spectrum derived with time series at certain grid point. Here the climatological condition is represented by two data sources, one with the CFSv2 data of the entire year 2016 (green curves) over the same domain, and the other as the wavenumber spectrum from Gage and Nastrom (1986), the Gage-Näström spectrum (the black curve). The Gage-Näström spectrum was obtained from thousands of commercial airplane flight measurements, and it is often used to represent the climatological power spectrum in the troposphere. The Gage-Näström spectrum has also been verified by the theoretical work of Lindborg (1999) on general two-dimensional turbulence behavior. For the climatological condition, the energy level and spectral slope

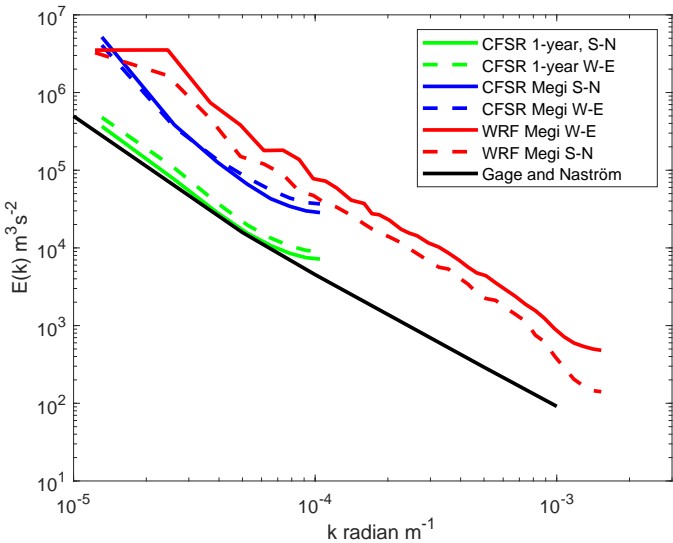

**Figure 2.** Power spectrum of wind speed over the space $E(k)$ as a function of wave number $k$, demonstrating higher spectral energy level associated with tropical cyclones in comparison with climatological conditions, with an example from Typhoon Megi. Here the $k-$spectra calculated, in the S-N and W-E directions, from the WRF modeling and from CFSv2 data during Typhoon Megi for 2016-09-26 10:00 to 2016-09-27 09:00 over water are compared with the spectra from one-year of CFSv2 data (2016) and the climatologically representative spectrum from Gage and Nastrom (1986).

are similar for the overlapping wave number range between the 1-year CFSR-1 data and the Gage-Näström spectrum, and both have significantly lower energy level than that for the Megi case. For this particular case, the higher resolution of the WRF model simulation, at 2 km, produced a spatial wind variability that is 3 - 5 times larger than that of the CFSv2 data at about 25 km resolution.

### 2.2.3 The SC-TC method

To adjust the SC method for more general storm conditions, we revise Eq. 3 to

$$S(f) = n \cdot a \cdot f^{-5/3} \tag{4}$$

with $n$ a coefficient reflecting weather types and its value can be adjusted with additional data, e.g. measurements. Thus for mid-latitude storms, $n = 1$, and for tropical cyclones, we develop a systematic way to determine $n$, which is described in the following. Here we need to decide how to link the new spectral tail (red dashed curve in Fig. 1) to the long term spectrum (the black curve). To keep it simple and to be on a relatively conservative side, we use the blue line to proceed with the calculation. When we are using Eq. 4, it is equivalent to generate a climatology with tropical cyclone ever-present during a year. Since we are only interested in the smoothing effect of the modeled time series on the once-a-year event, with rest of the data not considered, such an assumption is acceptable. Here, to match the IEC requirement, as also applied in Ott (2005), we correct

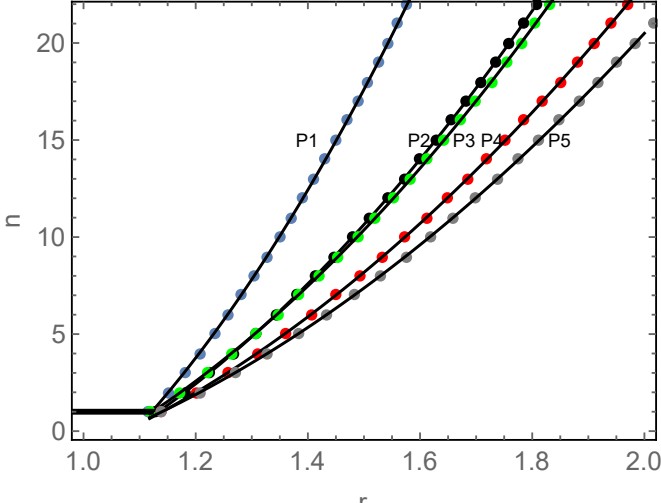

**Figure 3.** The relation between the enhance coefficient $n$ for the SC-TC method and the ratio between corrected and uncorrected 50-year wind using time series from five example grid points (Fig. 4). The dots are based on the calculation where $n$ was used as input varying from 1 to 30, and the curves are through regression based on the dots, following Eq. 5

the values to the equivalent 10-min resolution, hence $f_h = 72$ day$^{-1}$ is used when calculating the spectral moments using Eq. 3.

145     With an input of $n$, the SC-TC method would provide an output of the ratio of the corrected ($U_{50,SC-TC}$) and uncorrected ($U_{50,uncorr}$) value of the 50-year wind, $r$. By varying $n$ in Eq. 4 in connection with the use of the SC method, different $U_{50,SC-TC}$, and hence $r$, are obtained. Thus a relation between $n$ and $r$ can be derived. Such a relation depends on the initial wind speed time series and it therefore depends on the location of the interest in relation to the tropical cyclone structure, with $n$ increasing faster with $r$ for stronger wind region. This is demonstrated through the calculation from five grid points which are

150     marked in Fig. 4: P1: (17.5°N, 131°E); P2: (24°N, 130°E); P3: (26.5°N, 131°E); P4: (29°N, 133°E) and P5: (32.5°N, 136°E), and the corresponding value of $U_{50,BT}$ at the five grid points are 73, 67, 58, 53 and 45 m s$^{-1}$, respectively. P1 corresponds to the largest value over the domain. Here we used $n$ from 1 to 30 to obtain 30 corrected $U_{50,SC-TC}$ and hence $r$, and fit the data with second-order polynomial function. The corresponding $n - r$ relation is obtained as:

$$n = 1, \quad r \leq 1.12 \tag{5}$$

155
$$= \alpha r^2 + \beta r + c, \quad r > 1.12$$

For P1, $\alpha = 28.28$, $\beta = -30.24$ and $c = -0.66$. The 30 points and Eq. 5 are shown in Fig. 3 for P1. In the same figure, similar data for the other four grid points P2 to P5 are also plotted. It is clear that the $n - r$ relation is not uniform over the space. To be on the conservative side, we use the $n - r$ relation from P1. The uncertainty related to this choice will be discussed in section 3.

In finding $n$ for Eq. 4, here we derive $r$ as a function of a wind parameter that are globally available freely. Firstly, 32-year annual maximum winds $u_{max}$ are extracted from the CFSR-1 data from 1979 to 2010. The 50-year wind at its original resolution, $U_{50,uncorr}$, are calculated applying the Gumbel distribution to the 32 values with the Annual Maximum Method. Secondly, we use the results of the 50-year wind from Ott (2005), $U_{50,BT}$ (see section 2.1), to train the CFSR-1 data. We re-grid $U_{50,uncorr}$ (spatial resolution of about 40 km) to the grid points corresponding to $U_{50,BT}$ (spatial resolution of 1°) and calculate the ratio $U_{50,BT}/U_{50,uncorr}$. For the re-gridding, we simply find the value at the closest grid point without any interpolation. The values of the ratio cover a range from 1 to 2.6, as shown in Fig. 5a. Based on the scatter plot of the ratio with $U_{50,BT}$ in Fig. 5a, we can describe $r$ in a linear relation with $U_{50,BT}$ for different wind speed ranges. A linear regression is made for $27.5 < U_{50,BT} < 65$ m s$^{-1}$ following:

$$r_1 = a_1 U_{50,BT} + b_1 \tag{6}$$

where $a_1 = 0.0163$ and $b_1 = 0.569$. This relation is shown in Fig. 5a in red dashed line. We use a similar dependence of $r$ with uncorrected model value $u$ to Eq. 6:

$$r = a_2 u + b_2 \tag{7}$$

for $27.5 < u < 60$ m s$^{-1}$; $u$ is the uncorrected 50-year wind. Here we use $U_{50,BT}$ to calibrate Eq. 7. We use the same slope as Eq. 6, $a_2 = a_1$. By requiring the calculation of the 50-year wind through Eq. 4, $U_{50,corr}$ to match $U_{50,BT}$ at P1, we loop the calculation and obtain $b_2 = 0.62$. One constant of $r$, $a_2 \cdot 27.5 + b_2$, is then obtained for $U_{50,BT} < 27.5$ m s$^{-1}$ and another one, $a_2 \cdot 60 + b_2$, for an upper wind speed limit, e.g. $u > 60$ m s$^{-1}$. Thus:

$$\begin{aligned} r &= 1.07, \quad u < 27.5 \\ &= 0.0163u + 0.62, \quad 27.5 \leq u < 60 \\ &= 1.60, \quad u \geq 60. \end{aligned} \tag{8}$$

We stopped the loop at $n = 15$, when $U_{50,SC-TC} = 73.24$ m s$^{-1}$. Eq. 8 is plotted in Fig. 5b together with Eq. 6. We then use $r$ calculated from Eq. 8 as input to Eq. 5, to obtain $n$. For the initial, uncorrected 50-year wind less than 27.5 m s$^{-1}$, $n = 1$, which is the same as Eq. 3. Thus, the SC-TC method is ready to be used and it coincides with the SC method for $u < 27.5$ m s$^{-1}$.

## 2.3 Extreme wind at different heights

Usually, model wind outputs are provided at a few certain heights, e.g. 10 m or 100 m, and most models provide outputs at 10 m. The calculation of $U_{50}$ in Ott (2005) is at height $z = 10$ m, and the CFSR-1 data we used so far are also at 10 m. Modern offshore wind turbines can be as tall as 200 m. Therefore, we introduce a simple approach here to obtain winds at other heights $U_z$ from 10-m winds for storm conditions.

In extreme wind conditions, the logarithmic wind law can be considered valid up to several hundreds of meters according to Sonde wind speed measurements during hurricane conditions, as shown in e.g. Powell et al. (2003) and Giammanco et al.

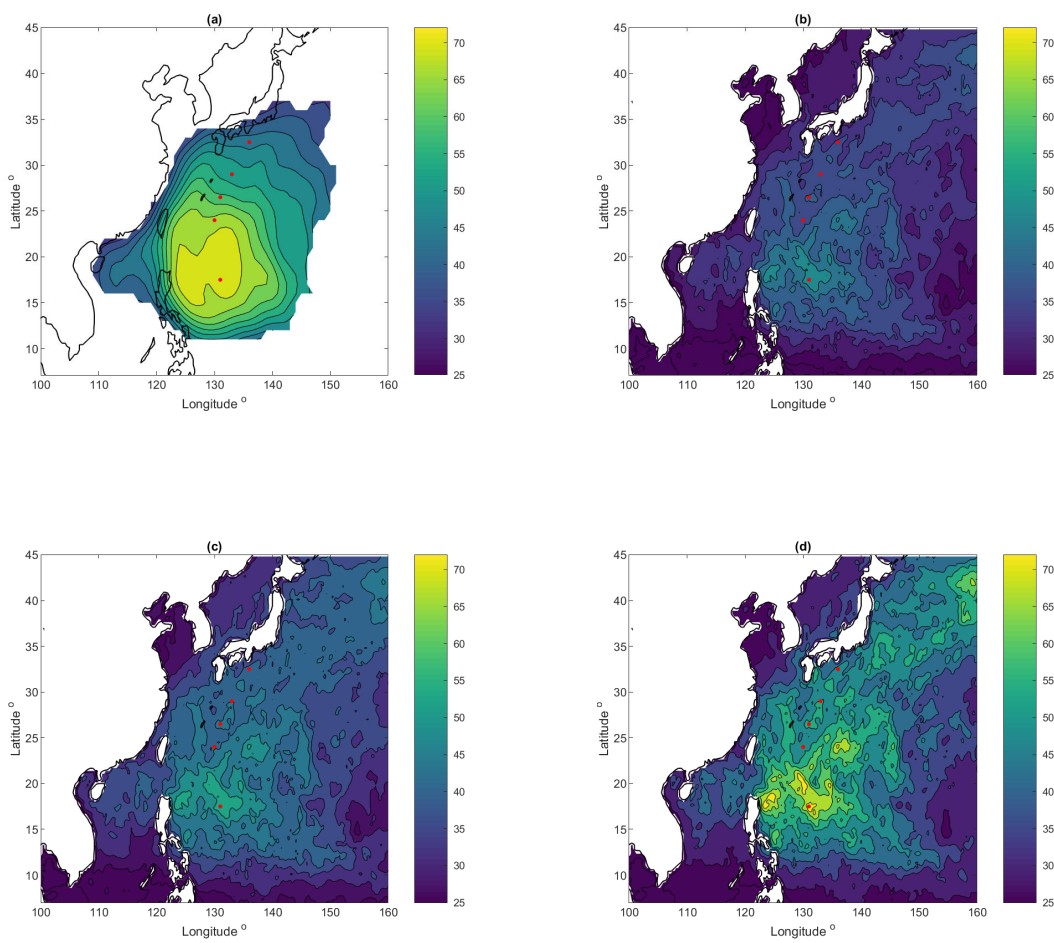

**Figure 4.** Spatial distribution of the 10-min 50-year wind at 10 m over southern North Pacific ocean. (a) re-produced with data (from Best Track and the Holland Model) from Fig. 13 in Ott (2005); (b) from the CFSR-1 data directly without spectral correction; (c) from CFSR-1 data with the spectral correction method (Eq. 3); (d) from CFSR-1 data with the SC-TC method developed here. The red dots mark the locations of five points: (17.5°N, 131°E), (24°N, 130°E), (26.5°N, 131°E), (29°N, 133°E) and (32.5°N, 136°E)

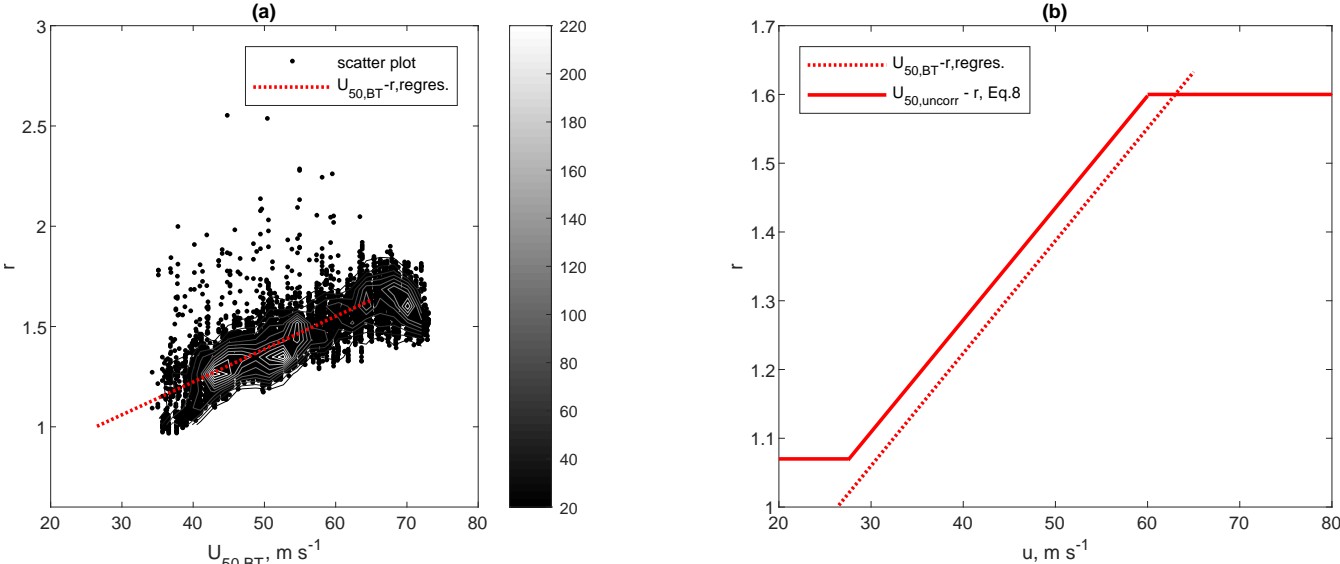

**Figure 5.** (a) Scatter plot of the ratio (the 50-year uncorrected wind from the CFSR-1 data and $U_{50,BT}$ from Ott (2005)) vs. $U_{50,BT}$, superimposed with the density contours with number of samples shown in the color bar. Also shown are the regression line from 27.5 to 65 m s$^{-1}$ with constants at the two ends; (b) The calibrated relation between $r$ and input $u$ (the uncorrected 50-year wind) in solid curves, together with the dashed curve from (a)

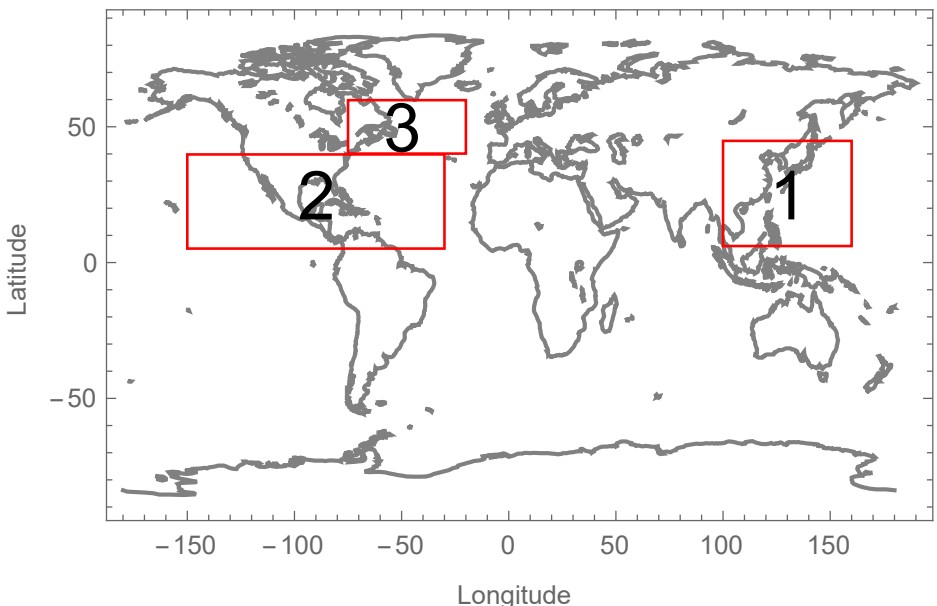

**Figure 6.** The three tropical cyclone affected regions chosen for the calculation of the extreme wind using the SC-TC method.

(2013). Thus we can calculate $U_z$ using

$$U_z = (u_*/\kappa)\ln(z/z_0). \tag{9}$$

where $u_*$ is the frictional velocity and $\kappa = 0.4$ is the von-Kármán constant. With $U_{10m}$ at $z = 10$ m known, $U_z$ can thus be obtained analytically if we have one more relation for $z_0$ and $u_*$.

There are several such relations established in the literature for describing $z_0$ and $u_*$. Here we examined three of them, with the first one the Charnock formulation for well-developed wind sea waves (Charnock, 1955) :

$$z_0 = \alpha_{ch}u_*^2/g, \tag{10}$$

where $\alpha_{ch}$ is the Charnock coefficient, which depends on the sea state, and it is larger for rougher sea. In numerical models, it is often between 0.011 to 0.02, as a function of wind speed. Here for the strong winds, we use $\alpha_{ch} = 0.02$.

The second one is the Andreas algorithm that takes into account of spray effect at strong wind conditions (Andreas et al., 2015):

$$u_* = 0.239 + 0.0433\left((U_{10m} - 8.271) + \left(0.12(U_{10m} - 8.271)^2 + 0.181\right)^{0.5}\right),$$

$$z_0 = 10\exp\left(-\kappa U_{10m}/u_*\right). \tag{11}$$

And the third are the algorithms that are used in wave model SWAN (e.g. Zijlema and van der Westhuysen, 2005). The SWAN algorithms are based on field measurements over a wide range of wind speed conditions, including hurricane conditions. It suggests a level-off or decreasing of surface drag coefficient ($C_d$) or roughness length with wind speed at strong winds, interpreted as a result from wave breaking processes:

$$C_d = \left(0.55 + 2.97(U_{10m}/31.5) - 1.49(U_{10m}/31.5)^2\right)10^{-3},$$

$$u_* = \sqrt{C_d}U_{10m},$$

$$z_0 = 10\exp\left(-\kappa U_{10m}/u_*\right). \tag{12}$$

In a test of calculating the wind speed at 100 m using wind speed at 10 m, it is found that the difference in using the three above-mentioned formulations is negligible for winds at 100 m up to 25 m s$^{-1}$. While the Charnock formulation with $\alpha_{ch} = 0.02$ gives very similar results to the Andreas formulation up to 40 m s$^{-1}$, it provides larger values for stronger winds, with a difference of about 1 m s$^{-1}$ at 50 m s$^{-1}$. Both the Charnock and the Andreas formulations give larger estimations than the SWAN formulations for winds larger than 25 m s$^{-1}$, and the overestimation increases to about 3 m s$^{-1}$ at 50 m s$^{-1}$. In this study we proceed our calculation using the SWAN formulation as it provides surface wave breaking effect at strong winds, which is calibrated with measurements (though very few) during hurricane conditions. We note that all of these formulations are over-simplification of the sea surface at such strong wind conditions during tropical cyclones and uncertainty is embedded.

## 3 Results and discussions

The SC-TC method is applied to three areas in this study and they are marked in Fig. 6 in boxes, with box-1 covering the same region as in Ott (2005) for typhoons and boxes 2 and 3 for hurricanes. These three areas are chosen as they host the most

severe and frequent tropical cyclones, based on the International Best Track Archive from National Centers for Environmental Information (NOAA) (NOAA, 2017).

For the three areas, the 50-year winds are calculated at 10 m, 50 m, 100 m and 150 m. With values at the four heights available, one can also obtain values at any height in-between using an interpolation or extrapolation method, e.g. polynomial. The data of these 50-year winds at the four heights are provided in the database (Larsén and Ott, 2022).

In order to show the effect of the spectral correction in comparison with the result from Ott (2005) (re-produced here as Fig. 4a), in Fig. 4, we plotted the 50-year wind at 10 m directly from the reanalysis data without spectral correction, $U_{50,uncorr}$ (Fig. 4b), the 50-year wind at 10 m using the spectral correction through Eq. 3 (Fig. 4c) and the 50-year wind at 10 m using the spectral correction for tropical cyclones through Eq. 4 (Fig. 4d). Fig. 4b, c and d share consistent wind distribution patterns. Compared to $U_{50,BT}$, the magnitude of the 50-year wind is significantly underestimated $U_{50,uncorr}$ in Fig. 4b (with the maximum value 50.8 m s$^{-1}$). The spectral correction method with Eq. 3 brings the maximum value 50.8 m s$^{-1}$ to 58.8 m s$^{-1}$ (Fig. 4c), which is still significantly smaller than the corresponding $U_{50,BT}$: 73.02 m s$^{-1}$. Comparing Fig. 4a and d, $U_{50,BT}$ vs. $U_{50,SC-TC}$, one can see that, first of all, the wind speed range is comparable for the common area of the two plots, being from about 35 to 73 m s$^{-1}$. Secondly, the locations of the strongest winds are consistent in the two plots, being between $120-140°$E and $15-25°$N. The contour lines in Fig. 4a are very smooth due to the use of Holland model and coarse resolution best track data (1°) in the Ott method, while Fig. 4b reveals richer spacial details.

When calculating $U_{50,SC-TC}$, the following three new equations have been used: Eq. 4, Eq. 5 and Eq. 8. Among them, the coefficients in Eq. 5 are derived with input of the model time series in connection of the use of the spectral correction method; we have used the time series from P1 for this. The coefficients in Eq. 8 have been calibrated using $U_{50,BT}$, so that $U_{50,SC-TC}$ matches $U_{50,BT}$ at P1. For both Eq. 5 and Eq. 8, we have used data from P1. By doing so, we are on the conservative side of the calculation and at the same time saved huge computational cost. Ideally such calibration should be done to all CFSR grid points. However, this can not be done due to the dis-match of grid size between $U_{50,BT}$ and the reanalysis data. Moreover, we do not have $U_{50,BT}$ outside the domain of Fig. 4a. This implies that it is inevitable that we will introduce assumptions for using Eq. 5 and Eq. 8 for most of the reanalysis grid points. Using the calibration from one point, P1, thus ensures a consistent workflow.

It is nevertheless relevant to address the uncertainty related to the use of calibration at one point. In theory, if we calibrate at all grid points using the same procedure as for P1, we get similar values of $U_{50,BT}$ and $U_{50,SC-TC}$ at these grid points. Thus, the difference between Fig. 4a and d will be able to directly suggest the related uncertainty. In section 2, we introduced five grid points (Fig. 4), whose $U_{50,BT}$ varies from about 45 to 73 m s$^{-1}$; the exact numbers are listed in Table 1. Among the 5 locations, the agreement between $U_{50,SC-TC}$ and $U_{50,BT}$ is good for P1, P3 and P4, while at P2, $U_{50,SC-TC}$ is still underestimated and at P5, $U_{50,SC-TC}$ is overestimated. We need to remember that the spatial distribution of $U_{50,BT}$ is of very coarse resolution, which can miss out spatial details and result in large uncertainties in some individual locations; P2 could be such a case. For the five points, we also calculated $U_{50}$ using Eq. 8 with the same coefficients as for P1, but together with their respective $n-r$ relations from the five points (Fig. 3). The values are also shown in Table 1. Here one can see that the estimates of the 50-year wind are underestimated.

**Table 1.** The estimation of $U_{50}$ (m s$^{-1}$) at the five points shown in Fig. 4 using Eq. 5 using their respective time series (ts) as well as using that from P1, in comparison with $U_{50,BT}$ and $U_{50,uncorr}$.

| $U_{50}$ | P1 | P2 | P3 | P4 | P5 |
|---|---|---|---|---|---|
| $U_{50,uncorr}$ | 50.67 | 39.61 | 39.72 | 38.47 | 38.47 |
| $U_{50,BT}$ | 73.02 | 66.75 | 57.80 | 53.41 | 44.66 |
| Eq. 5 with ts from P1 | 73.24 | 53.88 | 54.23 | 53.55 | 51.98 |
| Eq. 5 with ts from P2 | | 50.26 | | | |
| Eq. 5 with ts from P3 | | | 50.47 | | |
| Eq. 5 with ts from P4 | | | | 48.09 | |
| Eq. 5 with ts from P5 | | | | | 46.27 |

We point it out that in deriving the spectral correction method (section 2.2.1), we have assumed Gaussian process for the wind speed. As also pointed out and discussed in the original paper Larsén et al. (2012) that a Gaussian process is not ideal for describing the wind speed, it however makes it possible for us to derive analytically the effect on the once-in-a-year event in a time series that misses high frequency fluctuation. This also adds to the overall uncertainty.

The agreement between Fig. 4a and d is an encouragement to apply the SC-TC method to other tropical cyclone affected areas.

Figure 7 shows 10-min $U_{50}$ at 100 m, which is a more relevant height for modern offshore turbines, over the three areas as marked in Fig. 6. Note that the three boxes do not include all areas with the impact of tropical cyclones. Note also that the current study only addresses "water areas"; this is mainly because of two reasons. Firstly, extrapolation of surface winds to other heights over land requires much more complicated modeling approaches due to spatial inhomogeneous surface conditions. Moreover, it is not a trivial task to obtain such data for the surface conditions, whereas over water we benefit from the dependence of roughness length on the wind speed, even though there is uncertainty introduced. In addition, the best track data used for calibration in connection with the SC-TC method is mostly available over water.

When we extrapolate the wind speed from 10 m to higher altitudes, we have used the roughness length algorithms that are used in wave model SWAN, Eq. 12. These algorithms suggest that at hurricane strength, water surface roughness becomes smoother as a result of wave breaking and foaming processes. The corresponding roughness length is thus smaller than what the Charnock formulation Eq. 10 gives, and thus will provide smaller wind speeds at higher altitudes. One may argue if we should use again the more conservative approach such as the Charnock formulation. We choose not to use it because of the following reasons. Firstly, it has been proven incorrect at very strong winds. Secondly, Eq. 12 has been supported by measurements, even though very few. Thirdly, the difference caused by Eq. 10 and 12 is systematic; one can easily include this uncertainty in the assessment once there are further measurements for validation.

Even though we have used the CFSR-1 data in this paper, the SC-TC method can be applied similarly with other reanalysis data with sufficient data length. Though the quality of the reanalysis data, including the information of tropical cyclone parameters such as path, intensity and spatial structures, needs to be quality checked first. Imberger and Larsén (2022) show

that the characteristics of tropical cyclones are quite different in the following reanalysis data: CFSR-1, MERRA-2 (Modern-Era Retrospective analysis for Research and Applications), ERA-5 and CFDDA. For instance, among the four, the patterns of tropical cyclones are very weak in the CFDDA data. This suggests an important source of uncertainty associated with the input reanalysis data.

It should be pointed out that using the dependence of $n$ on the only one parameter, the wind speed, allows to use the SC-TC method in areas of strong winds that are not necessarily affected by tropical cyclones. It can be improved with further input of information when possible, to indicate if it is an area with tropical cyclone impact or not. In addition, the use of Eq. 4 has now been calibrated for mid-latitude storms and tropical cyclones, it is not yet examined for other types of extreme wind events such as thunderstorms; further studies are needed to extend the application of the method.

## 4   Summary and conclusions

This study develops a method for calculating the extreme wind for tropical cyclone affected areas, here called the SC-TC method. This is done by adjusting the spectral correction (SC) method from Larsén et al. (2012), through adding an enhancement coefficient to the spectral model, which is a function of the local extreme wind. Such a dependence is calibrated using the estimates from Ott (2005) who used the best track data and Holland model for the area of southern North Pacific ocean.

Here we summarize the main steps of the recipe for this method, which can be used in connection with other data sources, for instance, a new atlas that is based on measurements, and/or new reanalysis data:

1. Calculate $U_{50}$ from the original model time series over the domain: $U_{50,uncorr}$.

2. Find out the maximum value of $U_{50}$ from $U_{50,BT}$ and the corresponding location ($p_{0a}$). We call this maximum value $U_{50,BT,max}$.

3. Find out the maximum value of $U_{50}$ from $U_{50,uncorr}$ and the corresponding location ($p_{0b}$). In our study, $p_{0a}$ and $p_{0b}$ are very close to each other and we used $p_{0b}$ which is P1.

4. Extract model time series from $p_{0b}$ and apply the spectral correction method with $n$ ranging from e.g. 1 to 30 to obtain the corrected values (we call it $U_{50,test}$ here). Derive a relation between $n$ and $U_{50,test}/U_{50,uncorr}$, thus Eq. 5 can be defined.

5. Derive Eq. 8 from the scatter plot of $U_{50,BT}/U_{50,uncorr}$ vs. $U_{50,BT}$, and find the coefficients through looping the calculations until $U_{50,test}$ becomes close enough to $U_{50,BT,max}$.

The three equations Eq. 4, Eq. 5 and Eq. 8 are thus established. With an input $u$ as the uncorrected 50-year wind at 10 m from a given location, $r$ can be calculated using Eq. 8. With $r$ as input to Eq. 5, we obtain $n$, which can now be used in the SC-TC method using Eq. 4.

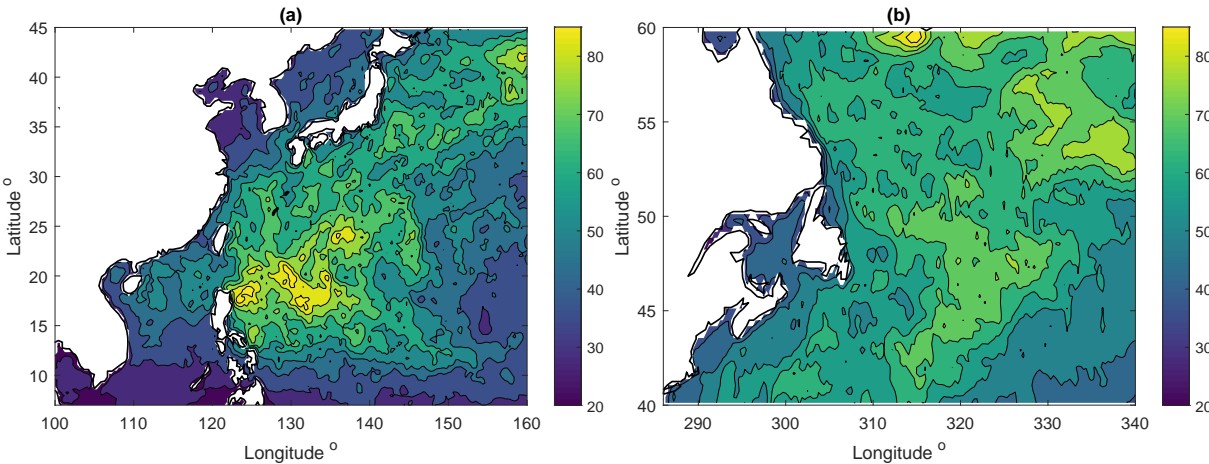

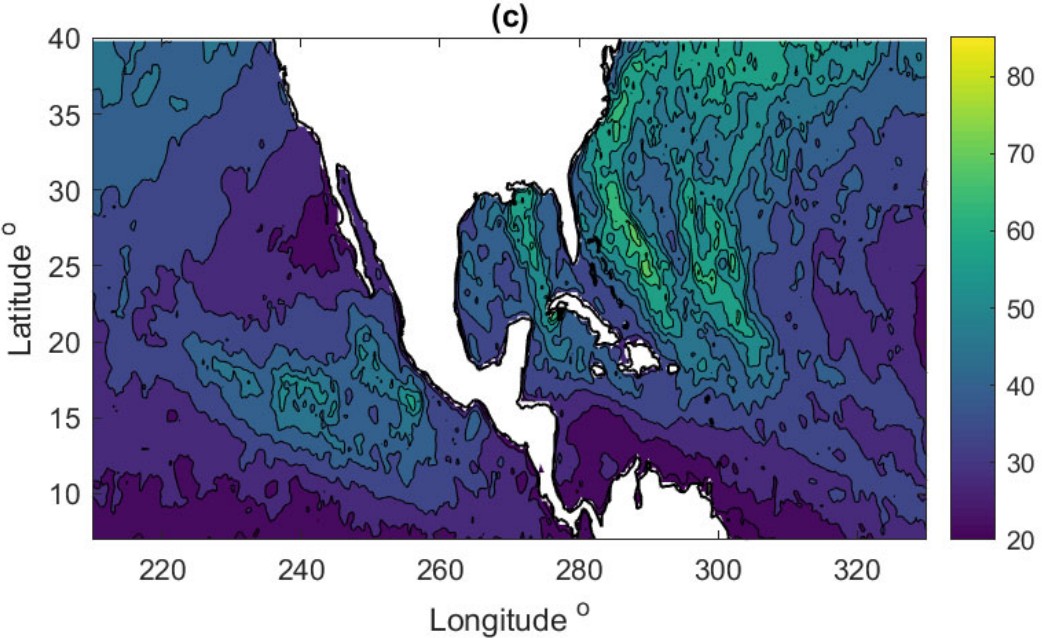

**Figure 7.** The 50-year wind of 10-min values at 100 m over water in three zones as in Fig. 6.

The results from the SC-TC method provide consistent wind distribution of 10-min 50-year wind at 10 m, and in addition much richer and more realistic spatial details, in comparison with the results from Ott (2005). The method can be applied for all water areas with tropical cyclones and can be used to obtain extreme winds from surface to a few hundreds of meters.

We acknowledge that extension of the SC method to the SC-TC method is quite empirical and there might be other approaches for improving the calculation. In spite of the uncertainties, the method and the data produced in this study serve to fill in the void of extreme wind estimations in most tropical cyclone affected areas. We acknowledge that we do not have enough measurements for validation, other than the study of Ott (2005), and therefore encourage validation from measurement owners.

*Data availability.* The CFSR data are publicly available at https://rda.ucar.edu/. The 50-year wind values over the three areas at 10 m, 50 m, 100 m and 150 m are available at https://zenodo.org/record/7089426.Yym676RBxZc, doi: 10.5281/zenodo.7089426

## Appendix A: Mesoscale modeling of typhoon Megi using WRF

There are two purposes for using the mesoscale Weather Research and Forecasting (WRF) model. The first is to examine the spectral energy level during an example case of typhoon in comparison of climatological conditions. The second is to compare the spectral energy level from the CFSv2 data (with a spatial resolution of tens of kilometers) with that from the WRF data (with a spatial resolution of 2 km).

We used WRF version 4.0, configured with the moving nest function. Three nested domains are used, with spatial resolutions of 18 km, 6 km and 2 km, respectively. The innermost domain has 339 by 342 grid points. We used 52 vertical model levels from the surface to a pressure level of 5000 Pa. We used the new Thompson microphysics scheme (Thompson et al., 2004), the RRTMG scheme for long and short wave radiation physics scheme (Iacono et al., 2008), MYNN 3.0 PBL scheme (Nakanishi and Niino, 2009) and Noah Land Surface Model. The Kain-Fritsch cumulus scheme (Kain and Fritsch, 1993) was used for the outer domain (I) but not for domains II and III. We used ERA5 data as initial and boundary conditions for WRF. The daily 0.25° OISST data were used to define the sea surface temperature conditions. The simulation started at 26th Sep. 2016 00:00 and ended at 27th Sep. 2016 12:00, with the first 12 hours as spinning-up period. The model outputs are recorded every 10 min.

Figure A1 shows the typhoon track from three data sources, the best track data, the CFSv2 data as the position of the lowest mean sea level pressure (MSLP) and the WRF data, also as the position of the lowest MSLP. They are in good agreement.

Both the wind speed data from CFRv2 and WRF from 12:00 on 26th to 12:00 on 27th over the innermost model domain are used to obtain the mean spectra of the longitudinal and meridional wind as shown in Fig. 2.

*Author contributions.* XL outlined the paper, developed the SC-TC model and did the calculation. SO provided data from Ott (2005) and suggestions. XL wrote the paper with suggestions from SO.

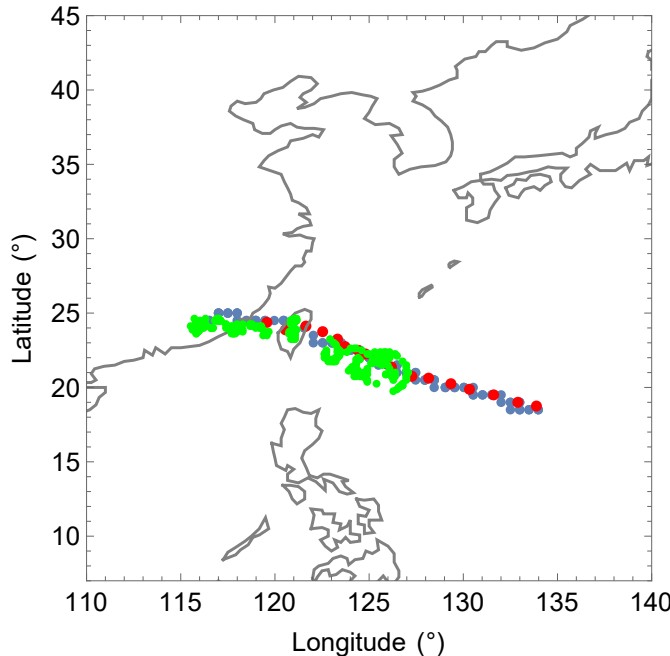

**Figure A1.** Center positions (with lowest mean sea level pressure) and track of Typhoon Megi in September 2016, red: best track data (23rd - 26th Sep. 2016); blue: CFSv2 data (23rd - 27th Sep. 2016); green: WRF data (26rd - 27th Sep. 2016).

*Competing interests.* The authors declare that they have no conflict of interest.

340 *Acknowledgements.* This study is supported by the EUDP projects GASP (EUDP J. nr. 64018-0095) and GASPOC (EUDP J. nr. 65020-1043). The CFSR products are downloaded from https://rda.ucar.edu/.

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
