# Peer review of "Adjusted spectral correction method for calculating extreme winds"

_Wind Energy Science, 2022_

## Referee Comment (RC1)

The authors developed a method to calculate the extreme wind for tropical cyclone affected water areas and the results are in agreement with Ott (2005) over their study domain. The paper is an interesting work as this method can be applied for all water areas with tropical cyclones and be used to obtain extreme winds from surface to a few hundreds of meters. However, some detailed information are missed in the manuscript, which makes it hard for reading. Therefore, I recommend minor revision including the implementation of the points and comments below.

(1) In page 6 line 139, it says that " we use the n-r relation from the grid point (17N, 130E) ", it is highly suggested to plot this point in Figure 3a.

(2) The curve of the relationship between $u_{max}$ and r at the grid point (17N, 130E) should be added, so the source of the Eq. (6) can be clear to the readers. Meanwhile, the annual wind maximum umax in Eq. (6) refers to the uncorrected annual maximum wind extracted from the CFSR data?

(3) In page 7 line 150, it says " Fig. 4b shows the inter-relationship between n, r and u.", however, the red curve line in Fig. 4b only refers to the dependence of n on r. Meanwhile, the caption of Fig 4b "The derived dependence of n on r as in Eq.4" should be modified as "The derived dependence of n on r as in Eq.5".

(4) The meaning of k and its value in Eq. (7), Eq. (9) and Eq. (10) should be added. Is it different from $\alpha_{ch}$ in Eq. (8)?

(5) Please check the description in page 10 line 216, "the use of Eq. 4 has…" or "the use of Eq. 5 has …"?

---

## Author Comment (AC1)

The authors are sincerely grateful for the valuable comments and suggestions from the reviewers. These comments and suggestions brought interesting discussion, added useful references, helped clarifying the concepts and improving the presentation of the paper. We give our response (in black) to these comments (in blue) in the following, point-by-point.

Reviewer 1

The authors developed a method to calculate the extreme wind for tropical cyclone affected water areas and the results are in agreement with Ott (2005) over their study domain. The paper is an interesting work as this method can be applied for all water areas with tropical cyclones and be used to obtain extreme winds from surface to a few hundreds of meters. However, some detailed information are missed in the manuscript, which makes it hard for reading. Therefore, I recommend minor revision including the implementation of the points and comments below.

(1) In page 6 line 139, it says that " we use the n-r relation from the grid point (17N, 130E) ", it is highly suggested to plot this point in Figure 3a.

Good point. The point is now added to this figure, which is the new Figure 4a. As we are informed not to upload the new version of the paper, we make a copy of the figure here. We also added four more points, which are used for discussion of uncertainties.

[Figure]

Figure 4. Spatial distribution of the 10-min 50-year wind at 10 m over southern North Pacific ocean. (a) re-produced with data from Fig. 13 in Ott (2005); (b) from the CFSR-1 data directly without spectral correction; (c) from CFSR-1 data with the spectral correction method (Eq. 3); (d) from CFSR-1 data with the SC-TC method developed here. The red dots mark the locations of five points: (17.5°N, 131°E), (24°N, 130°E), (26.5°N, 131°E), (29°N, 133°E) and (32.5°N, 136°E)

(2) The curve of the relationship between $u_{max}$ and r at the grid point (17N, 130E) should be added, so the source of the Eq. (6) can be clear to the readers. Meanwhile, the annual wind maximum umax in Eq. (6) refers to the uncorrected annual maximum wind extracted from the CFSR data?

The curve of the relationship is now added to this figure (now Figure 5).

[Figure]

**Figure 5.** (a) Scatter plot of the ratio of the 50-year wind from the CFSR-1 data and $U_{50,BT}$ from Ott (2005), with $U_{50,BT}$ and the density contours with number of samples shown in the color bar. Also shown are the regression line from 27.5 to 65 m s$^{-1}$ with constants at the two ends; (b) The calibrated relation between $r$ and input $u$, here the uncorrected 50-year wind (solid curve) and the dashed curve is from (a).

(3) In page 7 line 150, it says " Fig. 4b shows the inter-relationship between n, r and u.", however, the red curve line in Fig. 4b only refers to the dependence of n on r. Meanwhile, the caption of Fig 4b "The derived dependence of n on r as in Eq.4" should be modified as "The derived dependence of n on r as in Eq.5".

Thanks for pointing it out. The sentence has been re-written. Indeed it is Eq. 5 that should be in the sentence; it is now corrected.

(4) The meaning of k and its value in Eq. (7), Eq. (9) and Eq. (10) should be added. Is it different from $\alpha_{ch}$ in Eq. (8)?

k, or rather κ, is the von-Kármán constant; it is now added to the text. $\alpha_{ch}$ is, as already explained in the text, the Charnock coefficient.

(5) Please check the description in page 10 line 216, "the use of Eq. 4 has…" or "the use of Eq. 5 has …"?

It should be Eq. 4 in this sentence.

Review 2

Interesting method extending to tropical cyclone areas the use of the spectral correction method for mesoscale or reanalysis data. The study is limited to data from a rather old study from Ott (2005) which highlights the lack of benchmark data in these conditions. This motivates the authors to stay on the conservative side and adopt a simple correction factor (n) on the previous method for good reasons. The approach is robust and practical. The only thing I'm missing is **some quantification of uncertainty**, at least with regards to the spatial variability which is something that Ott data can provide. This would give more confidence when you extrapolate to the other large areas without validation data.

Thanks very much for such a thorough review with many good comments and suggestions.

1) Indeed, when deriving the relationships between r, n and u (eqs. 5 and 6), you mention that this could be site dependent but you end up using the location with the stronger winds to be on the conservative side. However, when you extrapolate to other cyclone areas you end up applying the same relationship at every grid point over very large areas so we may anticipate large errors normally biased to overprediction of the extreme winds. It would be interesting to apply the regression technique locally to each of the Ott (2005) grid points to compare with your results using eqs 5 and 6. This way you can provide some quantification of the potential bias/uncertainty introduced by your conservative approach at least for the validation area.

This is a very valid argument. And it is a good idea to provide an uncertainty assessment on this subject.

We added in section 3 "Results and discussions", paragraphs 4 and 5, to address this subject. The text reads:

When calculating $U_{50,SC-TC}$, the following three new equations have been used: Eq. 4, Eq. 5 and Eq. 8. Among them, the coefficients in Eq. 5 are derived with input of the model time series in connection of the use of the spectral correction method; we have used the time series from P1 for this. The coefficients in Eq. 8 have been calibrated using $U_{50,BT}$, so that $U_{50,SC-TC}$ matches $U_{50,BT}$ at P1. For both Eq. 5 and Eq. 8, we have used P1. By doing so, we are on the conservative side of the calculation and at the same time saved huge computational cost. Ideally such calibration should be done to all CFSR grid points. However, this can not be done due to the dis-match of grid size between $U_{50,BT}$ and the reanalysis data. Moreover, we do not have $U_{50,BT}$ outside the domain of Fig. 4a. This suggests that it is inevitable that we will introduce assumptions for using Eq. 5 and Eq. 8 for most of the reanalysis grid points. Using the calibration from one point, P1, thus ensures a consistent workflow.

It is nevertheless relevant to address the uncertainty related to the use of calibration at one point. In theory, if we calibrate at all grid points using the same procedure as for P1, we get similar values of $U_{50,BT}$ and $U_{50,SC-TC}$ at these grid points. Thus, the difference between Fig. 4a and d will be able to directly suggest the related uncertainty. In section 2, we introduced five grid points (Fig. 4), whose $U_{50,BT}$ varies from about 45 to 73 m s$^{-1}$; the exact numbers are listed in Table 1. Among the 5 locations, the agreement between $U_{50,SC-TC}$ and $U_{50,BT}$ is good for P1, P3 and P4, while at P2, $U_{50,SC-TC}$ is still underestimated and at P5, $U_{50,SC-TC}$ is overestimated. We need to remember that the spatial distribution of $U_{50,BT}$ is of very coarse resolution, which can miss out spatial details and result in large uncertainties in some individual locations; P2 could be such a case. For the five points, we also calculated $U_{50}$ using Eq. 8 with the same coefficients, but together the respective $n-r$ relations from the five points, as shown earlier in Fig. 3. The values are also shown in Table 1. Here one can see that the estimates of the 50-year wind are underestimated.

In addition, instead of all grid points, we chose 5 points (in the new Figure 4 – see my reply to reviewer-1) whose values from the best track data are from about 45 to 73 m/s for the relevant calculations of the uncertainty. New figures (Figure 3 & 4) and Table 1 are also made in connection with this investigation. Figure 4 has been copied in my reply to reviewer-1; Figure 3 and Table 1 are copied here:

[Figure]

**Figure 3.** The relation between the enhance coefficient $n$ for the SC-TC method and the ratio between corrected and uncorrected 50-year wind using time series from five example grid points (Fig. 4). The dots are based on the calculation where $n$ was used as input varying from 1 to 30, and the curves are through regression based on the dots, following Eq. 5

**Table 1.** The estimation of $U_{50}$ (m s$^{-1}$) at the five points shown in Fig. 4 using Eq. 5 using their respective time series (ts) as well as using that from P1, in comparison with $U_{50,BT}$ and $U_{50,uncorr}$.

| $U_{50}$ | P1 | P2 | P3 | P4 | P5 |
|---|---|---|---|---|---|
| $U_{50,uncorr}$ | 50.67 | 39.61 | 39.72 | 38.47 | 38.47 |
| $U_{50,BT}$ | 73.02 | 66.75 | 57.80 | 53.41 | 44.66 |
| Eq. 5 with ts from P1 | 73.24 | 53.88 | 54.23 | 53.55 | 51.98 |
| Eq. 5 with ts from P2 | | 50.26 | | | |
| Eq. 5 with ts from P3 | | | 50.47 | | |
| Eq. 5 with ts from P4 | | | | 48.09 | |
| Eq. 5 with ts from P5 | | | | | 46.27 |

2) Given the high uncertainty of the **vertical extrapolation** methods why not sticking to the more conservative and simpler Charnock method, which has confluence with Andreas equation up to 40 m/s? In fact, seeing the parabolic dependency of n with wind speed in Figure 4, one could argue that the n coefficient may be a **generalization of the Charnock constant for cyclone wind events**. Maybe you can try to relate the two?

Even though both approaches (Charnock and enhance coefficient in Eq. 5 derived from P1) suggest positive dependence on wind speed, and provide conservative estimates, they are two separate concepts. Although one may try yet another approach to train such a relation between the enhance coefficient *n* and the Charnock parameter, it is not needed in the approach proposed in this study. We added a paragraph in section 3 "Results and discussions" (the first paragraph on page 13) to address the choice of the roughness length algorithms:

When we extrapolate the wind speed from 10 m to higher altitudes, we have used the roughness length algorithms that are used in wave model SWAN, Eq. 12. These algorithms suggest that at hurricane strength, water surface roughness becomes smoother as a result of wave breaking and foaming processes. The corresponding roughness length is thus smaller than what the Charnock formulation Eq. 10 gives, and thus will provide smaller wind speeds at higher altitudes. One may argue if we should use again the more conservative approach such as the Charnock formulation. We choose not to use it because of the following reasons. Firstly, it has been proven incorrect at very strong winds. Secondly, Eq. 12 have been supported by measurements, even though very few. Thirdly, the difference caused by Eq. 10 and 12 is systematic; one can easily include this uncertainty in the assessment once there are further measurements for validation.

3) The paper has value as it is so it could be published after some editing but it would be more solid if these suggestions are addressed.

The **new recipe** is described in the paragraph of p5.128. I think it would be easier to follow if you use a numbered list with all the steps from input to output.

The whole text regarding the new recipe has been re-written, from page 6 to 8. In addition, the numbered list of the recipe is provided at the end of the paper in section "Summary and conclusions":

Here we summarize the main steps of the recipe for this method, which can be used in connection with other data sources, for instance, a new atlas that is based on measurements, and/or new reanalysis data:

1. Calculate $U_{50}$ from the original model time series over the domain: $U_{50,uncorr}$.

2. Find out the maximum value of $U_{50}$ from $U_{50,BT}$ and the corresponding location ($p_{0a}$). We call this maximum value $U_{50,BT,max}$.

3. Find out the maximum value of $U_{50}$ from $U_{50,uncorr}$ and the corresponding location ($p_{0b}$). In our study, $p_{0a}$ and $p_{0b}$ are very close to each other and we used $p_{0b}$ which is P1.

4. Extract model time series from $p_{0b}$ and apply the spectral correction method with $n$ ranging from e.g. 1 to 30 to obtain the corrected values (we call it $U_{50,test}$ here). Derive a relation between $n$ and $U_{50,test}/U_{50,uncorr}$, thus Eq. 5 can be defined.

5. Derive Eq. 8 from the scatter plot of $U_{50,BT}/U_{50,uncorr}$ vs. $U_{50,BT}$, and find the coefficients through looping the calculations until $U_{50,test}$ becomes close enough to $U_{50,BT,max}$.

**Some editorial corrections/suggestions:**

P1.21: particularly for over water > particularly over water

Suggestion taken.

P2.36: for the assessing > for assessing

Suggestion taken.

P2.37: are forecast > are forecasts

Suggestion taken.

P2.38: improvement > improvements

Suggestion taken.

P2.44: are of spatial > come with a spatial

Suggestion taken.

P2.45: has used > have used; extreme wind > extreme winds; data validation > validation

Suggestion taken.

P2.48: suffers from the smoothing effect…> suffer from the smoothing effect, introduced by a coarse grid that facilitates the convergence of the model.

Suggestion taken.

P2.53: wind variability to… > wind variability from modeled time series through a spectral model. Thus, the corrected time series will follow the power spectrum down to the temporal resolution of the measurements.

Suggestion taken.

P2.56: create extreme wind > predict extreme winds

Suggestion taken.

P3.59: same way as for the mid-latitude > same way for mid-latitude

Suggestion taken.

P3.62: the strenth of the methods from Ott (2005) and Larsen et al (2012).

Suggestion taken.

P3.75: which were from the best > which is derived from best track data, and Holland model, for an area over…

Suggestion taken.

P3.76: in contour lines > as contour lines

Suggestion taken.

P3.80: and, accordingly, translates into low spectral energy at high frequencies when compared to measurements.

Suggestion taken.

P3.85: zeroth- and second-order

Suggestion taken.

P4.110: from Gage and Nastrom (1986) for the same area? (please specify)

This sentence has now be re-written: "The Gage-Näström spectrum was obtained from measurements from thousands of commercial airplane flights, which is often used to represent the climatological power spectra in the troposphere. The Gage-Näström spectrum has also been verified by the theoretical work of \cite{lindborg.99} for general two-dimensional turbulence behavior."

P4.118: WRF-SWAN > have you coupled the SWAN model? This is not described in the Annex

It is a mistake – it is only WRF that is used. The text has now been corrected.

P4.119: the higher resolution of the WRF model simulation, at 2 km, produced a spatial wind variability that is 3 to 5 times larger than that of the CFSv2 data at 25 km resolution.

Suggestion taken.

P5.128: To define n in Eq.4, firstly, annual wind maxima u_max are extracted for a period of 32 years from 1979 to 2010 from CFSR-1 data. Then, the 50-year…

This part of text has been re-written.

P6.Fig2: calculated, in the S-N and W-E directions, from the WRF

Suggestions taken.

P6.131: to train the CFSR-1 data > to derive a regression model for the CFSR-1 data. (avoid machine learning jargon)

This part of text has been re-written.

P6.133: in grey dots, covering a range from 1 to 2.6.

This part of text has been re-written.

P7.Eq(6): replace u by u_max.

This part of text has been re-written.

P7.149: merges > coincides

Suggestion taken.

P7.154: a couple of hundreds of meters. We > 200 m. Therefore, we

Suggestion taken.

P7.157: to Sonde … conditions, as shown in e.g. Powell et al. (2003) and Giammanco et al. (2013)

Suggestion taken.

P8.Fig4: (a) Distribution… with respect to U_50ott; (b) Derived…

New figures have been made to replace the old Fig 4, with new text.

P9.167: algorithm

Suggestion taken.

P9.181: the difference > a difference

Suggestion taken.

P10:190: move the link to the references section

Suggestion taken.

P10.193: move the zenodo link to the references section using doi citation (thanks for sharing!)

Suggestion taken.

P10.203: Imberger and Larsén (2022) show

Suggestion taken.

P10.206: This suggest an important source of uncertainty associated to the input reanalysis data.

Suggestion taken.

P10.214: on only one parameter, the wind speed, allows to use the the SC-TC method in areas of strong winds that are not necessarily affected by tropical cyclones. (Is this what you mean?)

Yes that is what I mean. The text has been revised as suggested.

P12.234: two purposes for using

Suggestion taken.

P12.240: from the surface to a pressure level

Suggestion taken.

P12.243: as initial and boundary conditions for WRF.

Suggestion taken.

P12.244: OISST data were used to define the sea surface temperature conditions.

Suggestion taken.

P12.244: started… and ended at

Suggestion taken.

P12.245: The model outputs are recorded every 10 min.

Suggestion taken.

Reviewer 3

Referee comment on "Adjusted spectral correction method for calculating extreme winds in tropical cyclone affected water areas" by Xiaoli Larsén and Søren Ott, Wind Energ. Sci. Discuss., https://doi.org/10.5194/wes-2022-64-RC3, 2022
Adjusted spectral correction method for calculating extreme winds in tropical cyclone affected water areas – REVIEW

The article of Larsen and Ott addresses an interesting and highly relevant topic in the field of site assessment for wind energy. The authors present a promising approach towards a reliable extreme wind estimation from reanalysis data in regions affected by tropical cyclones.

However, a few points described in the following definitely need to be addressed before publication.

We thank the reviewer for such a thorough read and for the many detailed and good comments and suggestions.

1) The method development in section 2 particularly 2.2 lacks clarity and needs to be revised taking into account the following comments and questions.

The whole section 2.2 has been re-structured and re-written.

2) Line 84: "The maximum wind that occurs once a year …" is not precise in my opinion. In the framework of a Poisson process of wind velocities exceeding a threshold, Eq. (1) gives the velocity which is on average exceeded once in a period of T0. At this point there is no maximum estimation involved… However, there is a relation to the annual maximum but to be precise, it is not the same thing.

These sentences have been re-written:
"

The effect of the smoothing of the time series on the estimation of the extreme wind is calculated in Larsén et al. (2012) by assuming the time series follows a Gaussian process. The once-per-year exceedance can thus be described using the Poisson distribution. Following this, in Larsén et al. (2012), the wind that occurs once a year $\overline{U}_{max}$ was derived as a function of the zeroth- and second-order spectral moments $m_0$ and $m_2$:
"

Indeed at this point there is no maximum estimation involved, this step is for the calculation of the smoothing effect on the once-in-a-year event $U_{max}$.

3) Generally, the description of the SC method is pretty confusing to me. As far as I remember, the SC method calculates a correction factor sometime called smoothing effect as the ratio of Eq (1) for corrected and uncorrected spectra. This Factor is used to correct the the 50-year wind estimates of the reanalysis data, esimated by e.g. annual maximum method. Eq(1) is thus not directly used for estimating U50. O am I wrong?

In my experience a direct estimation with Eq (1) fails due to the various unfulfilled

assumptions (Gaussian wind, …).

We have now re-written this part and hope it becomes clear what the SC method does. This Factor can be used to correct the annual wind maxima samples, as well as the 50-year wind - the difference is actually negligible, which can be derived. Here we use the SC method to correct the annual wind maxima. The reviewer is absolutely correct that Eq. 1 is not used for estimating U50, it is used to obtain the correction factor.

We are aware of that a Gaussian process is not ideal for describing wind. It however makes it possible for us to analytically derive the impact on the once-in-a-year value caused by the missing energy in the spectral tail. This has been addressed in the original paper Larsén et al. (2011). The discussion is now also added in the new version.

4) Line 101-119: This is an interesting paragraph illustrating the effect of a typhoon on the spectra. But as far as I can see, it does not directly contribute to the SC-TC method. Thus the paragraph could be put into an extra section?!

Good point. We now break section 2.2 into three subsections and put the text (Line 101-119 from the original paper) into an extra section. This indeed improved the readability.

5) Line 125: Do you really need to mention version 2 of the enhanced spectrum?

Good point. It is not necessary to mention version 2. We now removed version 2 from the paper.

6) Line 131: "We regrid…" How do you regrid exactly? By bilinear interpolation?

A new sentence is now added to the text to explain the re-gridding: "For the re-gridding, we simply find the value at the closest grid point without any interpolation".

7) Line 134 and following: From here on, I get pretty confused. This might be also due to a lack of my expertise since some years have passed since I last worked at similar topics. However, since I am not completely new to the topic I should be able to follow you explanation of the method more easily.
Do you match the SC-TC results by tuning n to match the U50,Ott wind?
Is there on r for every grid point? If so, how could you get a relation for like EQ 5 for every grid point? How exactly do you estimate alpha and beta? Why do you choose a quadratic dependence of n on r?

I get that you show r dependent on U50,Ott in figure 4a. But how do you get a dependence on the annual maxima? Does this lead to a different r and n every year? I thank the authors in advance for clarifications.

The whole section has been re-written, with consideration of the reviewer's questions and comments. In addition, we made new figures (Fig. 3 & 5) to help the clarification, where Fig. 3 can be found in this reply-letter on page 4 (to reviewer-2) and Fig. 5 can be found here on page 2 (to reviewer-1). And we hope the new text explains all the above questions clearly. Similar comments were also raised by reviewer 2, who also suggested that we provide a numbered list of

applying this method. We followed that suggestion and made such a list in the last section "Summary and conclusions", which can be found in this reply-letter to reviewer-2 on page 5.

8) What is u exactly in Eq(6)?

The text has been re-written and u is the uncorrected 50-year wind at 10 m.

9) Last but not least, the conclusions in section 4 are no conclusions in my opinion. The section is just a slightly rephrased version of the abstract. However, the authors do offer some conclusions in the discussion. This might also be a matter of style nowadays. But I like conclusions to be conclusions not abstracts :)

Fair point!

We moved some 'conclusion' from section 3 "Results and Discussions" to the Conclusion section.

At the same time, we also revised the section title to "Summary and conclusion" as we find it a good suggestion from reviewer 2 to provide a numbered list of the method, and it seems like a good place to put it.

10) Other minor corrections have been mentioned in other comments already.

We have worked on those comments already and made changes accordingly.